# Study on the Mechanism of Grafting to Improve the Tolerance of Pepper to Low Temperature

**Huijun Long** [1,2] 🅞, **Ziyu Li** [1,2], **Huan Suo** [1,2], **Lijun Ou** [1,2], **Wu Miao** [1,2] **and Wenqiao Deng** [3,*]

1   College of Horticulture, Hunan Agricultural University, Changsha 410128, China
2   Huangpu Innovation Research Institute, Hunan Agricultural University, Guangzhou 510000, China
3   Changsha Academy of Agricultural Sciences, Changsha 410026, China
*   Correspondence: dengwenqiao2023@163.com; Tel.: +86-073184259495

**Abstract:** Pepper is a horticultural crop that does not tolerate low temperatures. To investigate how the grafted pepper responds to low temperature stress in the short term, transcriptome analysis was performed on grafted seedlings treated with low temperature for 1 h, 4 h, 12 h and 24 h compared with those treated for 0 h. The results showed that genes related to CAM4, MPK8, RbohD and OXI1 might be related to the response of grafted seedlings to low temperature stress in the short term. To investigate how low temperature tolerant rootstocks can improve the low temperature tolerance of grafted peppers, morphological and physiological indices of self-rooted and grafted seedlings were analyzed under low temperature conditions for different days. The results showed that the degree of wilting, REL and MDA content of grafted seedlings were significantly lower than those of self-rooted seedlings, and the antioxidant enzyme activities were significantly higher than those of self-rooted seedlings under low temperature stress. The results indicated that grafted pepper would activate ROS-related genes in a short period of time after low temperature stress and produce a large amount of ROS in response to the low temperature stress. When ROS accumulated to a certain level, the grafted pepper could increase the enzyme activity of antioxidant system to remove the ROS produced in the body, and help the pepper seedlings adapt to low temperature stress through osmoregulation mechanism, so as to resist the damage caused by low temperature. The results of the study provide ideas for growing pepper in low temperature environment.

**Keywords:** pepper; grafting; low-temperature tolerance; enzyme activity; osmotic substances; functional genes

## 1. Introduction

Pepper (*Capsicum annuum* L.), a warm and light-loving crop, is an annual herb of the genus Capsicum in the family Solanaceae, native to the tropics [1]. Low temperature is the main environmental factor limiting the growth and high quality and yield of pepper. During the seedling emergence period, continuous low temperature and rainy weather can easily cause rotten seeds or prolong the emergence time, resulting in low seedling emergence rate and uneven emergence, affecting the quality and later development of seedlings. During the seedling growing period, continuous low temperature and rainy weather can cause slow growth and yellowing of leaves; during the flowering and fruiting period, under continuous low temperature conditions, the tolerance of pepper is weakened and the absorption of nutrients is hindered, which can easily lead to diseases and anthocyanins, resulting in flower and fruit drop. During the fruiting period, the poor ability of pepper fruits to tolerate low temperature [2] will lead to frostbite, large spots on the fruit surface, production of rotten and deformed fruits, and low commercial value of the fruits. With global climate change, extreme weather frequently occurs in some areas of China, and extremely cold weather can affect winter pepper production. Improving the low-temperature tolerance of peppers is a fundamental measure to ensure sufficient supply of peppers in winter, which

also has important significance in achieving a balanced supply of peppers on an annual basis, and is also one of the powerful guarantees to increase the income of pepper farmers in winter and achieve rural revitalization.

At present, the main measures to improve the low temperature tolerance of peppers include planting low-temperature tolerant varieties [3], exercising low-temperature tolerance [4], spraying chemical biological agent [5,6], cultivation in facilities, and supplementing temperature when necessary. The complex topography and climate of China limit the cultivation of the few low temperature tolerant pepper varieties, while the use of chemical biological agent treatments and supplemental temperatures can increase production costs. In production, grafting is a viable means to improve the plants' ability to tolerate low temperatures [7–9]. Grafting can promote the stress tolerance [10,11], yield [12] and quality of peppers.

The cold resistance of plants is closely related to their antioxidant system. In response to low temperature stress, plants produce large amounts of reactive oxygen species (ROS), but the continued accumulation of ROS can lead to disruption of redox balance and oxidative damage [13]. At this time, plants will scavenge the reactive oxygen species and free radical ions produced in their bodies by producing antioxidant enzymes to counteract the damage caused by low temperature [14]. The content of malondialdehyde (MDA) increased significantly in sugar beet under 4 °C cold stress, the activity of antioxidant system enzymes such as T-AOC, SOD, POD, APX and CAT increased significantly, and the antioxidant capacity increased, thus removing the accumulation of reactive oxygen radicals and slowing down the damage caused by cell membrane lipid peroxidation to the cells [15]. After treating pepper with suitable concentration of inducer, on the one hand, it can increase SS content, PRO content and POD activity to relieve the damage caused by low temperature to pepper seedlings; on the other hand, it can reduce the relative conductivity and MDA content of leaves to maintain the balance of reactive oxygen metabolism system, reduce membrane lipid peroxidation, protect protoplasts from damage, and achieve the effect of enhancing the plant's resistance to stress [16]. Cucumber seedlings grafted with cold-tolerant rootstocks showed a significant increase in cold tolerance after low temperature treatment by decreasing electrolyte leakage rate and MDA content, increasing Pro content, and improving SOD, POD and CAT activities [17]. ROS metabolism changed significantly after 24 h of 8 °C treatment in pepper plants, and the activity of AsA, glutathione and NADPH dehydrogenase increased after 2 d and 3 d, indicating that these non-enzymatic metabolites and antioxidant enzymes can play a role in low temperature stress resistance through their effects on intracellular redox status [18].

In this paper, the wild aubergine S2 selected in the previous stage for its high tolerance to low temperature was used as rootstock, and the pepper autograft line TJ-A12 weak in tolerance to low temperature was used as scion. The early response of grafted seedlings to low temperature stress was investigated by comparing the transcriptome of grafted seedlings at 0, 1, 4, 12, and 24 h. By comparing the morphological and electrolyte permeability, MDA, Pro, SS, SOD, POD, CAT, APX, GR and other physiological indicators of self-rooted and grafted seedlings at 0, 1, 3, 5, 7, and 9 d after low temperature treatment, we aimed to elucidate the response of grafted peppers to low temperature and the response mechanism, and provide a practical basis for the application of grafting in early spring and late autumn facilities of peppers and the selection of varieties with low temperature tolerance.

## 2. Materials and Methods

### 2.1. Materials

The rootstock was the wild aubergine S2 with strong low temperature tolerance screened by the group in the early stage, and the pepper self-inbred line TJ-A12 with poor low temperature tolerance was used as the scion. Both were provided by the pepper group of the College of Horticulture, Hunan Agricultural University (Changsha, China).

### *2.2. Grafting and Low Temperature Treatment*

The seeds were selected from full-grained seeds, sterilized and soaked in hot-water treatment for 20 min, germinated in a constant temperature box at 28 °C, sown when the seeds turned white, and placed in an artificial climate chamber for further management. When the rootstock grew to being three-leafed with one bud and the scion was two-leafed with one bud, the seedlings were selected for cleft grafting [19]. After grafting, the normal management of water and fertilizer is based on the principle of "controlling branch growth to improve flower production and fruit yield". When the plants reached seven true leaves, grafted and self-rooted seedlings of similar growth and development were selected for a continuous low-temperature treatment at 4 °C. Other conditions were a photoperiod of 12 h day/12 h night, a light intensity of 500 $\mu M \cdot m^{-2} \cdot s^{-1}$ and 65% humidity. Each treatment was replicated three times with thirty plants in each replicate. The phenotypes were photographed at 0 day (d), 1 d, 3 d, 5 d, 7 d and 9 d of the low temperature treatment, and the corresponding leaves were taken at different time points for the determination of physiological parameters. Samples were taken from a mixture of three plants and replicated three times.

### *2.3. Comparison of Transcriptome Analysis of Self-Rooted and Grafted Seedlings*

#### 2.3.1. RNA Extraction, Library Preparation and Sequencing

Self-rooted and grafted seedlings treated at a low temperature at 4 °C for 0 h, 1 h, 4 h, 12 h and 24 h were taken, and total RNA was extracted from the leaf tissue using TRI reagent (Sigma Life Science, Santa Clara, CA, USA). Quality was checked by ribonuclease-free agarose gel electrophoresis to avoid possible degradation and contamination, and subsequently validated using an Agilent 2100 Bioanalyzer (Agilent Technologies, Santa Clara, CA, USA). Samples were sent to Biomarker Technologies (Beijing, China). for library construction and sequencing. After library construction, the libraries were tested for quality and then sequenced and analyzed using the Illumina platform.

#### 2.3.2. Processing of Sequencing Data

The raw data containing joints and low-quality Reads were filtered out to obtain high-quality Clean Data to improve the accuracy of the results. The Clean Reads were compared with the reference genome of pepper. DEGs were identified using R-packing edgeR and expression levels were calculated for each gene and normalized to FPKM. FDR was used to determine thresholds for *p*-values across multiple experiments. FDR < 0.05 and |log2 (Fold Change)| > 2 were used as significant cut-off points for differences in gene expression. DEGs were used for GO and KEGG enrichment analysis. GO terms with a corrected *p*-value < 0.05 and KEGG pathways with a *p*-value < 0.05 were considered significantly enriched.

#### 2.3.3. Functional Annotation

The GO "http://www.geneontology.org/ (accessed on 28 March 2022)". functional database and the KEGG "https://www.genome.jp/kegg/ (accessed on 28 March 2022)"pathway database were enriched for the differential gene set.

#### 2.3.4. Real-Time Fluorescence Quantitative PCR

Six genes identified by transcriptome analysis as significantly differentially expressed were randomly selected for detection by real-time fluorescence quantitative PCR (qPCR) technique. Actin was used as the internal reference gene, and each gene was repeated three times for the experiment. qPCR was performed using the SYBR Green chimeric fluorescence method provided by Vazyme Biotech Co., Ltd (Nanjing, China). with the primers shown in Table 1. The reaction system was 2×ChamQ Universal SYBR Qpcr Master Mix (10 μL), 0.4-μL Primer 1 (10 μM), 0.4-μL Primer2 (10 μM), 2-μL Tempalte DNA/cDNA, 7.2-μL ddH₂O. The reaction procedure was 95 °C for 30 s, 95 °C for 10 s, 60 °C for 30 s,

40 cycles in total. The $2^{-\Delta\Delta Ct}$ method was used to calculate the relatively high and low expression.

**Table 1.** Primer sequences used in qRT-PCR validation of transcriptome data.

| Gene ID | Forward Primer | Reverse Primer |
| --- | --- | --- |
| Capana02g003208 | GGTATGCGCTTGCCTCCCAT | TGGACCTCCTCGCCGAAAGA |
| Capana03g004005 | CGATGCTCCAAACAAGCGCA | GCCCAGCCCAAGTTACCCAG |
| Capana04g002884 | GTGGCTGCCAATGACGTTGC | GGAGTGGCTGGGGAGAGCTA |
| Capana07g000550 | AACAGGACATTGGCTGGGGC | TGCGACTCCCTTTGATGGCG |
| Capana09g000427 | TTCCGCGAGTCTCTCTGGCT | ATGACACCTCCTGCCCCCAT |
| Capana11g002375 | CCAAGGCGACCTGCCTCTTT | GTCGTCCCAAACTGCCCCAA |
| β-Actin | CCACCTCTTCACTCTCTGCTCT | ACTAGGAAAAACAGCCCTTGGT |

*2.4. Determination of Physiological Parameters*

Relative conductivity (REL) was determined by immersion method [20]; malondi-aldehyde (MDA) content was determined by thiobarbituric acid colorimetric method [21]; proline (PRO) content was determined by acid ninhydrin method [22]; soluble sugars (SS) were determined by anthrone colorimetric method [23]; superoxide dismutase (SOD) activity was determined by the nitrogen blue tetrazolium method [24]; peroxidase (POD) activity was determined by guaiacol method [25]; catalase activity (CAT) assay was performed according to the study of Trevor et al. [26]; ascorbic acid peroxidase (APX) activity assay was conducted using the ASA oxidation method [27]; glutathione reductase (GR) activity was determined by the DTNB method [28].

*2.5. Data Processing*

Microsoft office Excel 2010 was used for preliminary analysis of the data, SPSS 22.0 was used in ANOVA and Duncan's method was employed for multiple comparisons.

**3. Results**

*3.1. Transcriptome Analysis of Pepper Seedlings at Low Temperature*

3.1.1. Raw Data Collation, Filtering and Analysis

We sequenced the transcriptome of leaves from pepper self-rooted and grafted seedlings and obtained a large number of raw Reads in 30 samples. A total of 625,746,114 Clean Reads were obtained by quality checking to remove non-compliant sequences. The Clean Reads were matched to the pepper reference genome, and the effective matching rate was above 92.29% for all samples. After matching to the reference genome, 45,167 genes were identified, from which 34,466 genes were quantified, including 8871 novel genes. The quantified genes had 29,168 genes present in all samples, with 360, 193, 180, 317 and 396 specific genes at 0 h, 1 h, 4 h, 12 h and 24 h, respectively (Figure 1).

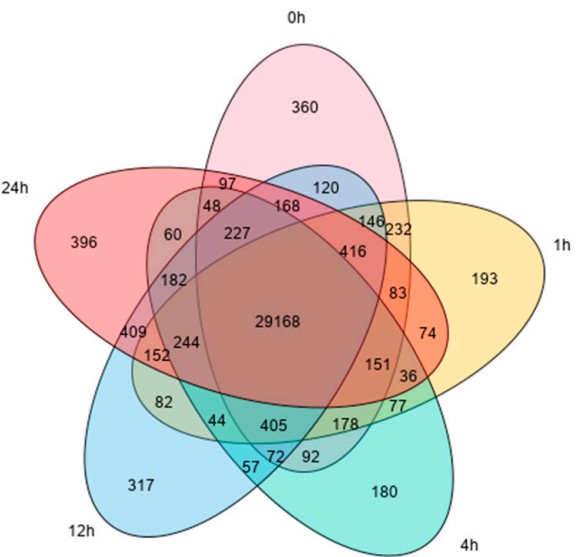

**Figure 1.** Venn diagram of genes in all samples.

### 3.1.2. Analysis of the Differentially Expressed Genes

To investigate the effects of different low temperature stress times on grafted seedlings, grafted seedlings treated with low temperature for 1 h, 4 h, 12 h, 24 h were compared with those treated with 0 h (labeled S0vsS1, S0vsS4, S0vsS12, S0vsS24, respectively) for transcriptome analysis. A total of 31,125 differentially expressed genes were identified in each comparison group, of which 4785 (up-regulated 2162, down-regulated 2623), 5496 (up-regulated 2572, down-regulated 2924), 8972 (up-regulated 4437, down-regulated 4535), and 11,872 (up-regulated 6282, down-regulated 5590) genes were obtained in the S0vsS1, S0vsS4, S0vsS12, and S0vsS24 groups, respectively (Table 2).

**Table 2.** Number of differential genes in grafted seedlings with different cold treatment times.

|  | **Up-Regulated** | **Down Regulated** | **Total DEGs** |
| --- | --- | --- | --- |
| S0 vs. S1 | 2162 | 2623 | 4785 |
| S0 vs. S4 | 2572 | 2924 | 5496 |
| S0 vs. S12 | 4437 | 4535 | 8972 |
| S0 vs. S24 | 6282 | 5590 | 11,872 |
| Total | 15,453 | 15,672 | 31,125 |

Note: S—Grafted seedlings. 0, 1, 4, 12, 24—The number of hours of low temperature treatment.

### 3.1.3. GO Analysis

In this paper, we performed GO enrichment analysis of differentially expressed genes in S0vsS1, S0vsS4, S0vsS12 and S0vsS24, respectively, and selected the top 10 functional entries with the most enriched differential gene entries in BP, CC and MF, respectively, to analyze the differences in changes of grafted seedlings under different low temperature treatment times. "Metabolic process", "cellular process", "single-organism process", "biological regulation", and "response to stimulus" are the five most enriched GO terms in the Bio-logical Process (BP) ontology. "Cell", "cell part", "membrane", "membrane part", and "organelle" are the five most enriched GO terms in the cellular component (CC) ontology. "Binding", "catalytic activity", "transporter activity", "nucleic acid binding transcription factor activity", and "molecule function regular" are the three most enriched GO terms in the Molecular Function (MF) ontology (Figure 2).

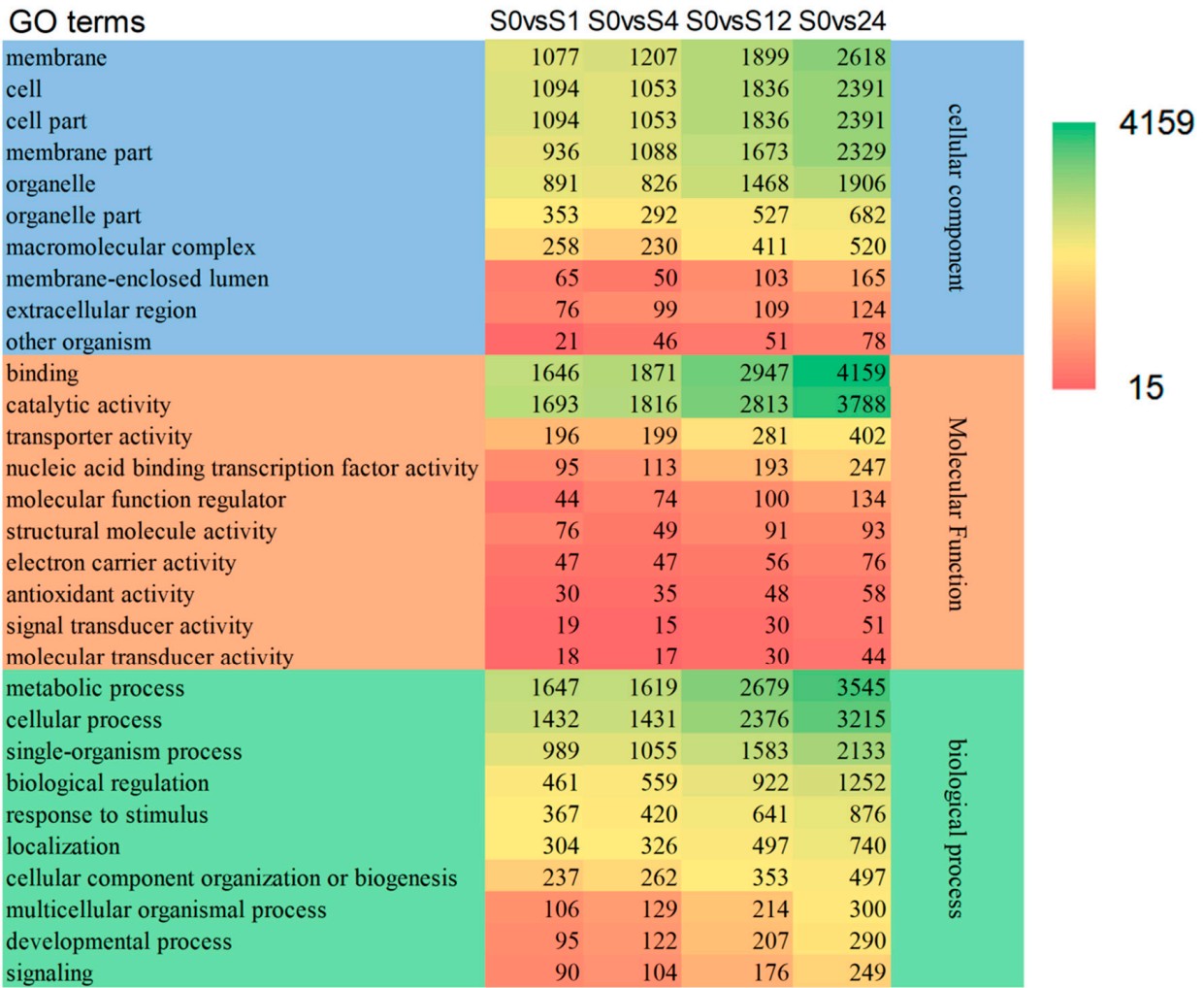

**Figure 2.** GO analysis of differentially expressed genes. Note: S—Grafted seedlings; 0, 1, 4, 12, 24—The number of hours of low temperature treatment.

3.1.4. KEGG Analysis

KEGG pathway reveals the function of differential genes that may be related to cold tolerance process in pepper. We selected the top 20 KEGG pathways with the highest differential gene enrichment for analysis. The results showed that the differentially expressed genes in S0vsS1, S0vsS4, S0vsS12 and S0vsS24 were mainly enriched in "Plant-pathogen interaction", "Plant hormone signal transduction", "MAPK signaling pathway-plant", "Starch and sucrose metabolism", "Protein processing in endoplasmic reticulum", and "Carbon metabolism". We found an interesting phenomenon that the longer the time of low temperature treatment, the more differentially expressed genes were enriched in these metabolic pathways, especially in "Plant hormone signal transduction","MAPK signaling pathway-plant", "Starch and sucrose metabolism ", the number of differential genes increased by 2.0, 1.7 and 0.7-fold, respectively. This suggests that grafted seedlings may respond to low temperature stress by regulating metabolism under low temperature treatment (Figure 3).

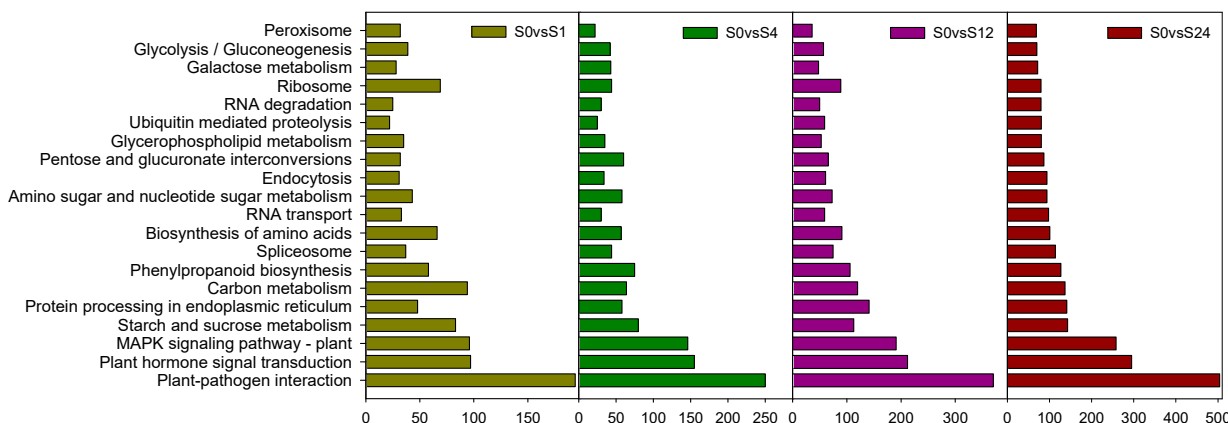

**Figure 3.** KEGG enrichment analysis of the differentially expressed genes. Note: S—Grafted seedlings; 0, 1, 4, 12, 24—The number of hours of low temperature treatment.

In the "MAPK signaling pathway-plant", we found that nine, two, five, and one genes each of calmodulin (CAM 4), mitogen-activated protein kinase 8 (MPK 8), respiratory burst oxidase (RbohD) and serine/threonine-protein kinase OXI1 (OXI 1) were significantly expressed at different low temperature treatment times of grafted seedlings. Among these genes, we found that the expression of most genes differed significantly in grafted seedlings, except for Capana01g000646, Capana11g002375, Capana06g001086, and most genes showed elevated expression after 12 h of low temperature treatment, with the highest expression of CAM4 (Capana03g001590 and Capana09g001938). The increase in expression was, in descending order, for OXI1, CAM4, RboD and MPK8, indicating that OXI1 and CAM4 may be closely related to the response of pepper to low temperature stress (Figure 4).

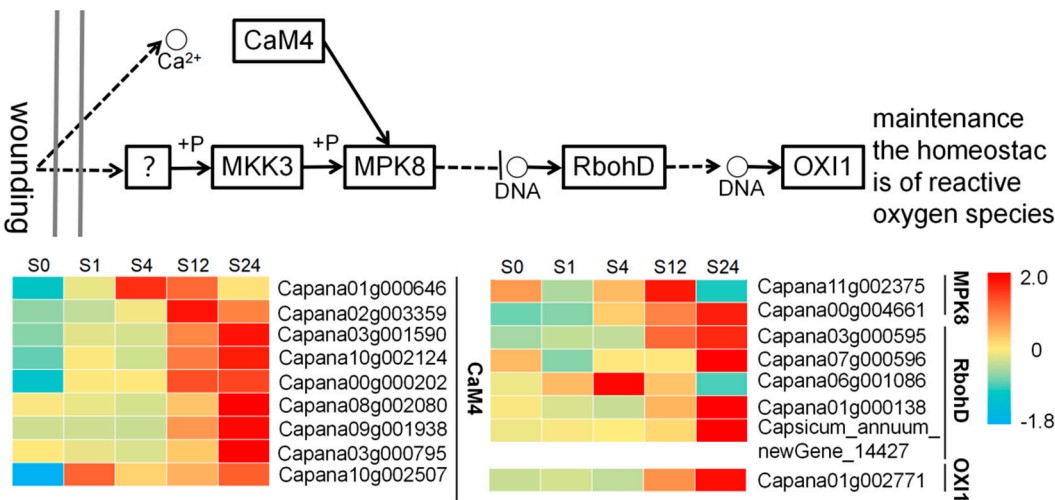

**Figure 4.** Comparative analysis of "MAPK signaling pathway-plant" genes.

### 3.1.5. qRT-PCR Validation

To verify the reliability of the transcriptome expression data, six genes identified as significantly differentially expressed by transcriptome analysis were randomly selected and subjected to fluorescent qRT-PCR validation. qRT-PCR results of the six differentially expressed genes showed similar trends to the RNA-seq sequencing results low-temperature treatment in both self-rooted and grafted seedlings at different time points (Figure 5).

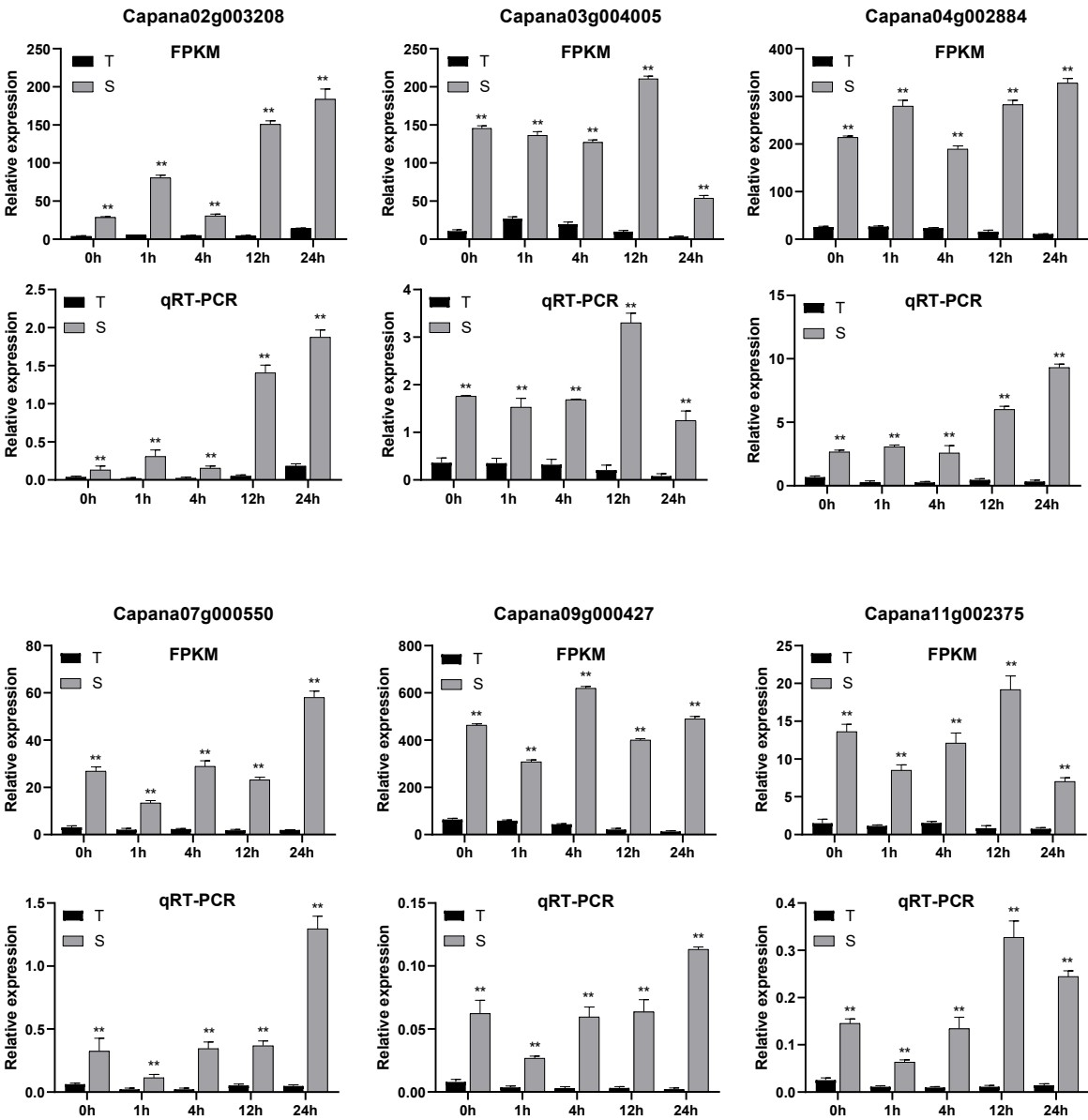

**Figure 5.** qRT-PCR validation of six differentially expressed genes. Note: S—Grafted seedlings; T—self-root seedlings; "**" means $p \leq 0.01$.

### 3.2. Effect of Low Temperature on Pepper Morphology

Under low-temperature conditions, the degree of leaf wilting gradually increased with treatment time, but there were large differences in the wilting phenomenon and the degree between self-rooted and grafted seedlings. On day 1, the leaves of the self-rooted seedlings wilted and drooped, while the grafted seedlings showed more obvious wilting until day 5. Nine days later, the leaves of the self-rooted seedlings showed severe water loss, wilting, drying, smaller leaf area and softer stems, while the grafted seedlings showed slight water loss at the leaf margins (Figure 6).

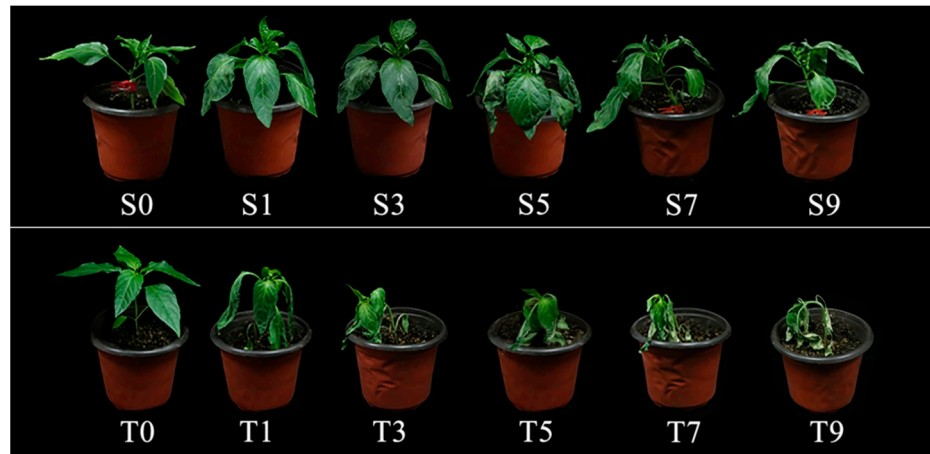

**Figure 6.** Morphological changes of pepper at different days of low temperature treatment. Note: S—Grafted seedlings; T—self-root seedlings; 0, 1, 3, 5, 7, 9—Number of days of low temperature treatment.

*3.3. Effect of Low Temperature on Physiological and Biochemical Indicators of Pepper*

3.3.1. Effect of Low Temperature on Osmoregulation in Pepper

Throughout the stress process, the electrolyte leakage rate of grafted seedlings was always lower than that of self-rooted seedlings. On day 1, the electrolyte leakage rate of self-rooted seedlings increased sharply, while that of grafted seedlings did not change much; on day 5, the electrolyte leakage rate of self-rooted seedlings was as high as 79.23%, while that of grafted seedlings was 34.83%; on day 9, the electrolyte leakage rate of self-rooted seedlings and grafted seedlings were 65.81% and 38.60%, respectively, which were 163.62% and 103.02% higher than those before the stress process, showing significant differences (Table 3). Under low-temperature stress, SS and Pro of both self-rooted and grafted seedlings showed an increasing trend, but there were significant differences in the degree of changes between the two. Before low-temperature stress, the SS content of self-rooted and grafted seedlings was not significantly different. On day 9, the SS content of grafted and self-rooted seedlings was 65.74 and 45.31 respectively, and the SS content of grafted seedlings was 1.5 times higher than that of self-rooted seedlings, which was significantly different (Figure 7A). On day 9, the Pro content of self-rooted and grafted seedlings increased by 48.72% and 57.50%, respectively, compared with that before stress, and the Pro content of grafted seedlings was 9.8% higher than that of self-rooted seedlings on day 9 (Figure 7B).

**Table 3.** Relative conductivity of grafted and self-rooted seedlings.

| Treating Days | T [1] | S [2] |
| --- | --- | --- |
| 0 | 24.97% ± 0.010 gh | 19.01% ± 0.015 i |
| 1 | 59.76% ± 0.031 d | 23.01% ± 0.020 h |
| 3 | 66.96% ± 0.006 bc | 38.89% ± 0.010 e |
| 5 | 79.23% ± 0.040 a | 34.83% ± 0.015 f |
| 7 | 69.18% ± 0.015 b | 27.63% ± 0.006 g |
| 9 | 65.81% ± 0.015 c | 38.60% ± 0.012 e |

[1] S—Grafted seedlings. [2] T—self-root seedlings. Note: Lowercase means significance at 0.05 levels. Means of 3 replications ± SE.

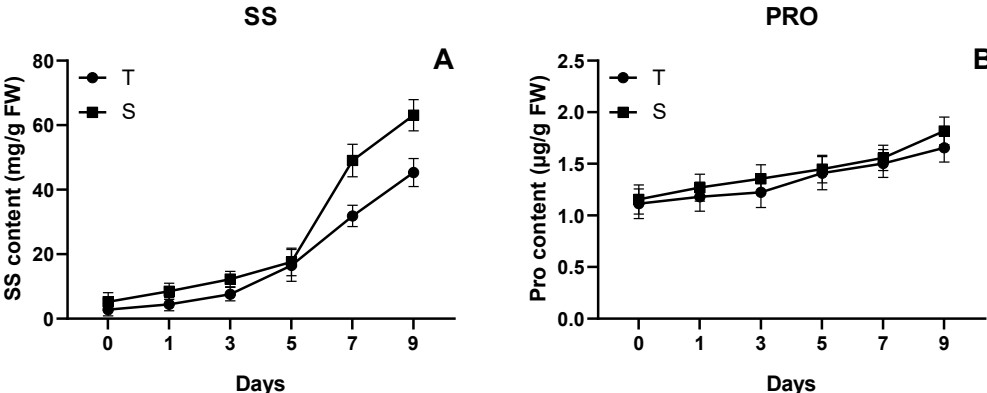

**Figure 7.** Soluble sugar (SS) (**A**), proline (PRO) (**B**) of grafted and self-root seedlings. Note: S—Grafted seedlings; T—self-root seedlings.

3.3.2. Effect of Low Temperature on Membrane Lipid Peroxidation in Pepper Seedlings

The MDA content of both self-rooted and grafted seedlings showed an increasing trend, but the increase of self-rooted seedlings was significantly higher than that of grafted seedlings. Before the low-temperature stress, the MDA content of self-rooted seedlings and grafted seedlings was 30.17 and 30.48 respectively. On day 5, the MDA content of self-rooted seedlings reached a peak (216.5824), while that of grafted seedlings was 33.97; on day 9, the MDA content of self-rooted seedlings was 379.99% higher than that before the stress, and that of grafted seedlings was 11.06% higher than that before the stress (Figure 8A). The SOD activity of both the self-rooted and grafted seedlings increased and then decreased, but the decrease was significantly greater in the self-rooted seedlings than in the grafted seedlings. On days 0 to 1 of the low-temperature treatment, SOD activity increased in both, with the SOD in grafted seedlings showing a higher increase than that in self-rooted seedlings. On day(s) 1 to 3, the SOD activity of both seedlings decreased significantly; after another 3 days, both self-rooted and grafted seedlings showed a gradually stable trend. On day 9, the SOD content of grafted seedlings was 27.32% higher than that of self-rooted seedlings, and grafted and self-rooted seedlings increased by 1.57% and 4.50%, respectively, compared to the pre-stress level (Figure 8B). The POD activity of grafted seedlings changed more actively than that of the self-rooted seedlings. On day 1, the POD activity of the self-rooted and grafted seedlings increased by 9.11% and 25.99%, respectively; on day 9, the POD activity of the grafted seedlings was 501.17% higher than that of the self-rooted seedlings with significant differences (Figure 8C). The CAT activity of self-rooted seedlings showed a 'rise–fall–rise' pattern, while that of grafted seedlings rose and then fell. Before the low-temperature treatment, the CAT activities of the self-rooted and grafted seedlings were 256.36 and 176.60 respectively; on day 3, the CAT activities of the two were very similar; on day 9, the CAT activity of the grafted seedlings was 47.72% higher than that of the self-rooted seedlings, showing a significant difference (Figure 8D). The pattern of change of APX activity in both self-rooted and grafted seedlings showed an increase followed by a decrease and then an increase again. The activity of CAT was 106.59% and 106.04% higher on day 9 than before the stress process in self-rooted and grafted seedlings, respectively, and was 56.95% higher in grafted seedlings than in self-rooted seedlings, showing a significant difference (Figure 8E). On day 9 of low-temperature stress, the GR activity of grafted seedlings was 1.7 times higher than that of self-rooted seedlings, with significant differences (Figure 8F).

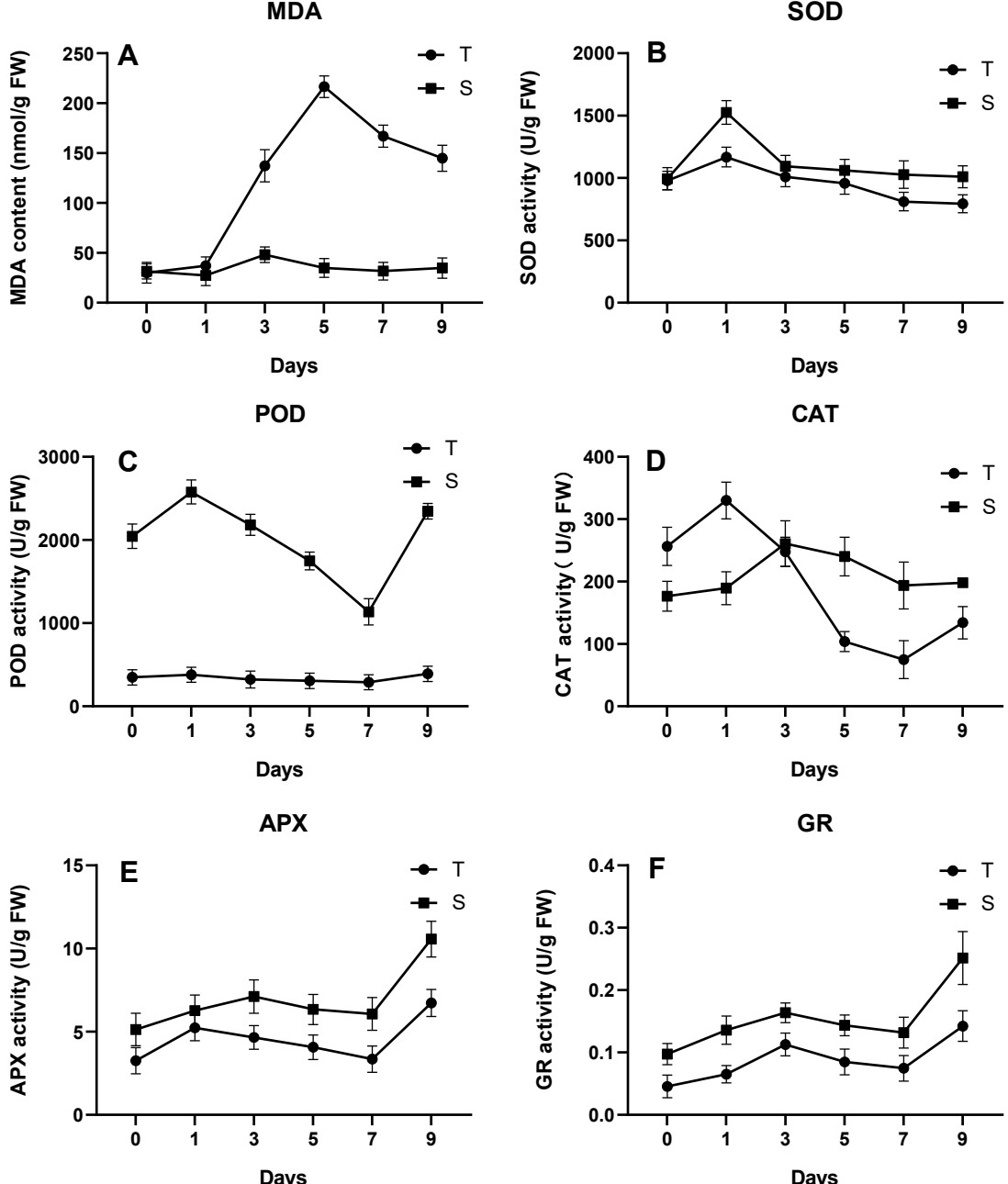

**Figure 8.** Malondialdehyde (MDA) (**A**), Superoxide dismutase (SOD) (**B**), peroxidase (POD) (**C**), catalase (CAT) (**D**), ascorbate peroxidase (APX) (**E**), Glutathione reductase (GR) (**F**) of grafted and self-root seedlings. Note: S—Grafted seedling; T—self-root seedling.

## 4. Discussion

### 4.1. Short-Term Response of Grafted Seedlings to Low Temperature

When plants are suddenly subjected to low temperature stress from suitable growth conditions, the cell membrane system is the first to be injured, and the wounding signal is generated when plants are subjected to biotic or abiotic stress. Studies have shown that $Ca^{2+}$/CaM signaling can rapidly recognize and transduce wounding signals to help plants adapt to environmental stresses [29]. In this paper, except for Capana01g000646, which was highly expressed at 4 h and then gradually decreased, most CAM4-related genes in grafted seedlings increased in expression after 12 h of low temperature treatment, and the expression increased significantly at 24 h, indicating that CAM4 can quickly respond to the damage caused by low temperature stress to grafted seedlings and help grafted

seedlings adapt to the low temperature environment. MAPK has an important role in plant resistance to abiotic stresses, and the transcript level of SbMAPK in Salicorniabrachiata increases dramatically under drought, low temperature, and high salt conditions, and the highest expression is observed under low temperature stress [30]. Overexpression of ZmSIMK1 in Arabidopsis enhanced its salt tolerance and induced the expression of stress-related genes RD29 and P5CS1 [31]. In addition, MAPK genes have a positive effect on flax response to saline stress [32]. Production of reactive oxygen species (ROS) is a conserved immune response primarily mediated by NADPH oxidases (NOXs), also known in plants as respiratory burst oxidase homologs (RBOHs) [33]. RBOHs play an important role in plant growth and development, adversity stress and cellular signal transduction mainly by regulating the production of reactive oxygen species, and the expression of RbohD gene tends to increase and then decrease in oilseed rape under drought, low temperature and high salt treatment compared with oilseed rape under suitable environmental conditions to regulate ROS production through the expression of Rboh gene [34]. It has been shown that after injury in Arabidopsis, the $Ca^{2+}$/CaM signaling pathway rapidly receives wounding signals and then activates mitogen-activated protein kinase 8 (MPK8), and the MPK8 pathway maintains oxidative homeostasis in plants by regulating the expression of the RbohD gene and thus influencing the accumulation of ROS [35]. In this paper, the expression of MPK8-related genes in grafted seedlings showed an increasing trend from 0 to 12 h of low temperature stress, and at 24 h, Capana11g002375 had a decreasing trend while Capana00g004661 still continued to increase, indicating that MPK8 also played a role in the resistance of grafted seedlings to low temperature stress. At the same time, the expression of RbohD in grafted seedlings started to increase at 12 h of low temperature treatment, while the expression of RbohD increased significantly at 24 h. This indicates that the cell membrane of grafted seedlings was injured at about 12 h of low temperature stress, and the expression of related genes increased after the grafted seedlings sensed the low temperature stress signal and produced ROS to resist the low temperature stress injury. However, ROS is a "double-edged sword" in the organism. On the one hand, ROS can play an important role as a signaling molecule in plant growth and development and stress response; on the other hand, the excessive accumulation of ROS can cause oxidative damage to biological macromolecules [36]. The wounding response that occurs minutes to hours after plant injury has two functions: first, to activate the defense response system and second, to promote healing of the injured tissue from further injury [37]. We predict that the wounding response occurring at 0–24 h of low temperature treatment of grafted seedlings belongs to the activation of the defense response system, and with the increase of low temperature stress time, the expression of MPK8-related genes will increase significantly after the accumulation of ROS to inhibit RbohD protein kinase, reduce the accumulation of ROS and maintain the oxidative balance in grafted seedlings, so as to reduce the damage of low temperature stress on grafted seedlings. Capana06g001086 in RbohD was highly expressed at 4 h, followed by a gradual decrease in expression, which may be the result of the indirect action of MPK8. Studies have shown that OXI1 also plays an important role in plant defense and root hair development [38], and the expression of OXI1-related genes in this paper gradually increased with the extension of low temperature treatment time, suggesting that OXI1 may play an important role in the resistance of grafted seedlings to cold damage.

### 4.2. Mechanism for Improving Low Temperature Tolerance of Grafted Seedlings

Osmoregulation is a protective regulatory process of the plant itself when faced with stress. In adversity, a large amount of osmoregulatory factors accumulate in the plant to increase the concentration of cytosol, which is used to respond to sudden danger signals, thereby achieving a protective effect on the plant. Leaf electrolyte leakage is an important indicator of plant cell permeability [39], and Pro and SS are important regulators whose levels change when subjected to stress [40–43]. In this paper, we found that the electrolyte permeability of both self-rooted and grafted seedlings of pepper increased significantly

compared with that before the stress process, and after the end of stress, the electrolyte permeability of self-rooted seedlings was significantly higher than that of grafted seedlings, indicating that grafting could effectively improve the cold resistance of pepper. In this study, we found that the Pro and SS contents of self-rooted pepper and grafted seedlings gradually increased with the increase of low temperature treatment days, and that the Pro and SS contents of grafted seedlings were significantly higher than those of self-rooted seedlings at each treatment stage. Grafting significantly increased the Pro and SS contents of pepper, which helped pepper seedlings adapt to low temperature stress through osmoregulatory mechanisms.

When plants are exposed to low temperature, the first to detect and receive a damage signal is the cell membrane, which cannot provide better protection for the plant if injured and would cause a series of physiological and biochemical reactions to cells such as increasing cell membrane permeability, disturbing the balance of reactive oxygen species and stimulating membrane lipid peroxidation, resulting in plant death in severe cases [44]. The content of MDA can indirectly reflect the degree of damage to the membrane system and the plant's ability to tolerate low temperature [45]. In this study, we found that the MDA content of both self-rooted and grafted seedlings increased overall compared with that before the stress, and that the change in MDA content of self-rooted seedlings was significantly greater than that of grafted seedlings, indicating that the increase in MDA content of pepper seedlings was effectively suppressed by grafting, which played a protective role.

When plants are subjected to low temperature stress, a large amount of ROS will be produced, which can participate in various metabolisms and is the second messenger regulating cell growth and death. In a normal growth environment, ROS should be in a state of dynamic equilibrium in the plant; under low temperature stress, the large amount of ROS production will cause a series of biochemical reactions such as DNA damage, protein inactivation, cell membrane peroxidation and even cell dysfunction and death [46]. This is when the plant's antioxidant enzyme scavenging system comes into play, showing appropriate stress response to regulate reactive oxygen species and free radical levels as a means of protecting the plant. SOD, POD, CAT, APX and GR are some of the main antioxidant enzymes in plants. SOD is mainly responsible for scavenging oxygen negative ions produced during temperature stress, while other enzymes are responsible for removing intracellular $H_2O_2$, thereby alleviating the damage caused by low-temperature stress to plants [47–50]. In this study, the antioxidant enzyme activities of pepper self-rooted and grafted seedlings showed an overall increasing trend, and the activities of POD, APX and GR all showed a significant increased, indicating that a certain period of low-temperature exercise could improve antioxidant enzyme activities in the plants. Moreover, the activities of SOD, POD, CAT, APX and GR of grafted seedlings were significantly higher than those of self-rooted seedlings at the end of low-temperature stress process. Grafting can increase the activity of antioxidant enzymes in pepper seedlings, effectively scavenge reactive oxygen species generated by low temperature, and maintain ionic balance, thus improving the cold tolerance of pepper.

## 5. Conclusions

Grafted pepper will activate ROS-related genes in a short period of time after low temperature stress and produce a large amount of ROS in response to the low temperature stress. When ROS accumulates to a certain level, grafted peppers can increase the enzymatic activity of antioxidant system to scavenge the ROS produced in the body and thus help pepper seedlings adapt to low temperature stress through osmoregulatory mechanism, thus resisting the damage caused by low temperature. This paper provides theoretical support for further research on both the mechanism of action of eggplant rootstock grafting to enhance the cold tolerance of pepper scions and on stress-resistant cultivation techniques. These results provide a theoretical basis and application value for promoting high quality cultivation of pepper under low temperature environment, high yield and low energy con-

sumption green production and selection and breeding of varieties with low temperature tolerance ability.

**Author Contributions:** Conceptualization, L.O. and W.D.; methodology, L.O. and W.D.; validation, H.L., Z.L. and H.S.; formal analysis, H.L. and Z.L.; investigation, Z.L. and H.L.; resources, W.M. and L.O.; data curation, Z.L. and H.L.; writing—original draft preparation, H.L. and Z.L.; writing—review and editing, W.D., W.M. and L.O.; visualization, H.L. and Z.L.; supervision, L.O., W.D., W.M. and H.S.; project administration, W.D.; funding acquisition, L.O. All authors have read and agreed to the published version of the manuscript.

**Funding:** This research was funded by the National Specialty Vegetable Industry Technology System Post Expert Funds and the Huangpu Research Institute Project of Hunan Agricultural University.

**Data Availability Statement:** Not applicable.

**Conflicts of Interest:** The authors declare no conflict of interest.

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
