# Peer review of "Study on the Mechanism of Grafting to Improve the Tolerance of Pepper to Low Temperature"

_agronomy, doi:10.3390/agronomy13051347_

Round 1
Reviewer 1 Report
Overview:
In the manuscript the authors aimed to elucidate the response of grafted peppers to low temperature and the response mechanism, by comparing the morphological and physiological indicators and gene expression differences between self-rooted and grafted seedlings after low-temperature treatment. The wild aubergine S2 selected for its high tolerance to low temperature was used as rootstock y, and the pepper autograft line TJ-A12 weak in tolerance to low temperature was used as scion. Overall, the paper adds novel knowledge about the role of grafting in the improved abiotic stress plant tolerance. The current study is on a topic of relevance and general interest to the readers of the journal. However, some important issues in the methodology and in the results interpretation and discussion need to be clarified and corrected. Therefore, a major revision process is necessary before the final acceptance.
Major comments:
1. The results regarding the plant morphology and the physiological and biochemical indicators is interesting. In fact, the grafting of a cold sensitive scion on a cold tolerant rootstock has a positive effect on cold tolerance and on plant physiological response to the stress. On the other hand the transcriptomic analysis is not properly designed and conducted, considering the aim of the work and the correlation of the gene expression data with the physiological and biochemical indicators. The time scale of the RNA sequencing (from 0 to 24 hours) is suitable for an early response analysis of plant to cold treatment and is not comparable with the time scale used for the physiological and biochemical indicators analysis (form 0 to 9 days). For this reason the two sets of data cannot be correlated, because representative of two distinct temporal phases of the response of the plant to cold stress. I suggest to use transcriptomic data concerning the same time points used for the physiological and biochemical indicators. If is not possible I suggest to consider the two sets of data as separated experiments and discuss them separately.
2. The comparisons between groups of the same time point used for the differential expression analysis (S0/T0, S1/T1, S4/T4 etc.) allow to identify genes whose expression is influenced by grafting and not by low temperature. This last information can be exploited by comparing the same sample (grafted or self-rooted) in different time of cold exposure, and could be interesting to obtain such information. For the same reason what is stated from line 236 to line 242 is not correct considering the comparisons studied and the volcano plot derived from these comparisons.
3. Neither in the results nor in the discussions are present information about the names and punctual functions of the genes resulted differentially expressed in the comparisons and involved in cold tolerance. There are only general information derived from GO and KEGG enrichment. The enrichment analysis returns information about function and pathways involved but no details about the differentially expressed genes. For this reason it cannot be concluded that “there is an increased expressions of genes related to the content of osmolytes, enzyme activities of antioxidant system etc.” (lines 383-387). No gene lists are present in the manuscript and no correlation can be supposed because of the different time course of the analysis (days with hours).
4. An important literature about plant response to abiotic stress exists and the strong involvement of the antioxidant and osmolyte systems is well documented. The genes involved in this kind of response are well-known and well-studied in many transcriptomic experiments (Sicilia A, Testa G, Santoro DF, Cosentino SL, Lo Piero AR. RNASeq analysis of giant cane reveals the leaf transcriptome dynamics under long-term salt stress. BMC Plant Biol. 2019;19(1):1–24; Sicilia A, Santoro DF, Testa G, Cosentino SL, Lo Piero AR, et al. Phytochemistry. 2020;177:112436). I think that the introduction and discussion sections could be improved by further describing this literature and the results could also take advantage by drawing up a list of tolerance candidate genes.
Minor comments
1. Introduction section: please provide more background about the correlation between abiotic stress (cold stress) and oxidative stress and antioxidant enzymes.
2. Introduction section, lines 59-61: please provide reference for the low temperature tolerance of the wild aubergine S2 rootstock.
3. Section 2.3: please indicate abbreviations of the physiological parameters and compound that are analyzed and described later in the text.
4. Section 3.3.2, lines 232-236: the number of DEGs obtained after applying the threshold to FDR and log2FC values (3507 total genes, 2228 up- and 1279 down-regulated) are different from those shown in Figure 5. Please check and correct.
5. Section 3.3.2, Figure 5: figures of low quality. Please improve the quality
6. Page 9: wrong numeration of paragraphs; paragraph 3.3.3 was skipped.
7. Section 3.3.4, Figure 8: figure of low quality. Please improve the quality. Please indicate in the figure caption which comparison the capital letters correspond to.
8. Section 4.3, lines 374-377: this sentence is a repetition of the sentence before.
Author Response
For more information, please refer to the attachment

Reviewer 2 Report
This study investigated the response and response mechanism of grafted peppers to low temperature. The authors compared the morphological and physiological indicators, as well as gene expression, between self-rooted and grafted pepper seedlings after exposure to low temperature. It is an interesting study. See specific comments below.
Line 80: What fertilizer did you use?
Line 85: What was the layout of your experiment?
Line 266: There is no 3.3.3
Line 286-288: Check the grammar of the sentence. Suggest adding “under” before “low temperature”.
Line 349-355: This is a very long sentence. Check similar throughout the manuscript.
Line 401: Make sure your format is consistent with the Journal-required format.
Author Response
Thank you for your comments, please see the attachment for details of the changes。

Round 2
Reviewer 1 Report
Thank you for the detailed response to the comments and I’m glad you have accepted my suggestions. Your explanations and the new structure in terms of methodology and results presentation helped to clarify some important aspects and to obtain more significant results, leading to an improvement of the quality of the manuscript. Just few little issues remained.
1. Please check if the abstract is in line with the journal instructions in terms of number of characters
2. Provide informations about the colour scale showed in figures 2 and 4. Use a legend or describe it in the figure caption
3. The conclusions should include the future perspetcives, the applications and the benefits to which the results obtained will lead.
4. Finally a last check for typos and formatting errors is necessary
Author Response
We feel great thanks for your professional review work on our article. As you are concerned, there are several problems that need to be addressed. According to your nice suggestions, we have made corrections to our previous draft the detailed corrections are listed in the annex.
